

# Medial knee cartilage is unlikely to withstand a lifetime of running without positive adaptation: a theoretical biomechanical model of failure phenomena

Ross H. Miller[1,2] and Rebecca L. Krupenevich[3]

[1] Department of Kinesiology, University of Maryland, College Park, MD, United States of America
[2] Neuroscience & Cognitive Science Program, University of Maryland, College Park, MD, United States of America
[3] Joint Department of Biomedical Engineering, University of North Carolina, Chapel Hill, NC, United States of America

Corresponding author
Ross H. Miller, rosshm@umd.edu

## ABSTRACT

Runners on average do not have a high risk of developing knee osteoarthritis, even though running places very high loads on the knee joint. Here we used gait analysis, musculoskeletal modeling, and a discrete-element model of knee contact mechanics to estimate strains of the medial knee cartilage in walking and running in 22 young adults (age 23 ± 3 years). A phenomenological model of cartilage damage, repair, and adaptation in response to these strains then estimated the failure probability of the medial knee cartilage over an adult lifespan (age 23–83 years) for 6 km/day of walking vs. walking and running 3 km/day each. With no running, by age 55 the cumulative probability of medial knee cartilage failure averaged 36% without repair and 13% with repair, similar to reports on incidence of knee osteoarthritis in non-obese adults with no knee injuries, but the probability for running was very high without repair or adaptation (98%) and remained high after including repair (95%). Adaptation of the cartilage compressive modulus, cartilage thickness, and the tibiofemoral bone congruence in response to running (+1.15 standard deviations of their baseline values) was necessary for the failure probability of walking and running 3 km/day each to equal the failure probability of walking 6 km/day. The model results suggest two conclusions for further testing: (i) unlike previous findings on the load per unit distance, damage per unit distance on the medial knee cartilage is greater in running vs. walking, refuting the "cumulative load" hypothesis for long-term joint health; (ii) medial knee cartilage is unlikely to withstand a lifetime of mechanical loading from running without a natural adaptation process, supporting the "cartilage conditioning" hypothesis for long-term joint health.

## INTRODUCTION

Knee osteoarthritis is the most common form of lower limb osteoarthritis and a major source of disability worldwide (*Vos et al., 2012*). Causal factors in the disease's initiation and progression are complex and contentious, but mechanical load on cartilage during gait appears to be an important factor. Once knee osteoarthritis has initiated, walking with relatively high peak loads on the knee is a risk factor for further progression of the disease (*Miyazaki et al., 2002*; *Chang et al., 2007*; *Bennell et al., 2011*; *Chehab et al., 2014*; *Chang et al., 2015*; *Hatfield, Stanish & Hubley-Kozey, 2015*; *Brisson et al., 2017*; *Davis et al., 2019*). Similar evidence on joint loading and initiation of knee osteoarthritis is sparse but appears in observational studies on older adults (*Amin et al., 2004*; *Lynn, Reid & Costigan, 2007*) and younger adults following knee surgery (*Hall et al., 2015*; *Teng et al., 2017*). Mechanical testing of articular cartilage explants suggests the fatigue life of cartilage is exhaustible within a number of loading cycles relevant to the human lifespan, at stress/strain levels well below the ultimate strength of the material (*Weightman, Freeman & Swanson, 1973*; *Weightman, Chappell & Jenkins, 1978*; *Chen et al., 1999*; *Bellucci & Seedhom, 2001*; *Sadeghi, Espino & Shepherd, 2017*; *Riemenschneider et al., 2019*). These tests are performed on cartilage explants that cannot heal/repair, although cartilage is often considered to lack substantial capacity for natural healing given its lack of direct innervation and vascularization in a healthy state.

These data suggest that an activity like running, which repeatedly places high peak loads on the knee, should increase the risk of developing knee osteoarthritis when performed frequently for appreciable periods of the human lifespan. However, recreational running is not associated with an increased risk of knee osteoarthritis (*Konradsen, Hansen & Sndergaard, 1990*; *Lane et al., 1998*; *Lo et al., 2018*) and may even be beneficial for knee joint health (*Van Ginckel et al., 2010*; *Horga et al., 2019*). *Miller (2017)* suggested three hypotheses to explain these observations. First, that contact mechanics cause cartilage strains in running to be unremarkable, again similar to those in walking. Second, stemming from the first hypothesis, that the damage accumulated on the knee cartilage from running is unremarkable, similar to the damage from walking the same distance. Third, that the mechanical loading of running triggers an adaptation in living cartilage that condition it to safely withstand these strains.

These hypotheses are difficult to test with standard approaches to estimating knee joint loading in biomechanics, which are based primarily on resultant knee joint moments from inverse dynamics and model-based estimates of joint contact forces. Related issues are the difficulty of observing the natural history of osteoarthritis, which can initiate and progress over many years, and the ethics of attempting to experimentally induce osteoarthritis. In such cases, computational models of tissue strain and cumulative damage can be useful for estimating fatigue life and failure probability over an extended period of time for tissues undergoing cyclical loading from locomotion (*Taylor, Casolari & Bignardi, 2004*; *Edwards et al., 2009*; *Landinez-Parra, Garzón-Alvarado & Vanegas-Acosta, 2011*; *Gardiner et al., 2016*).
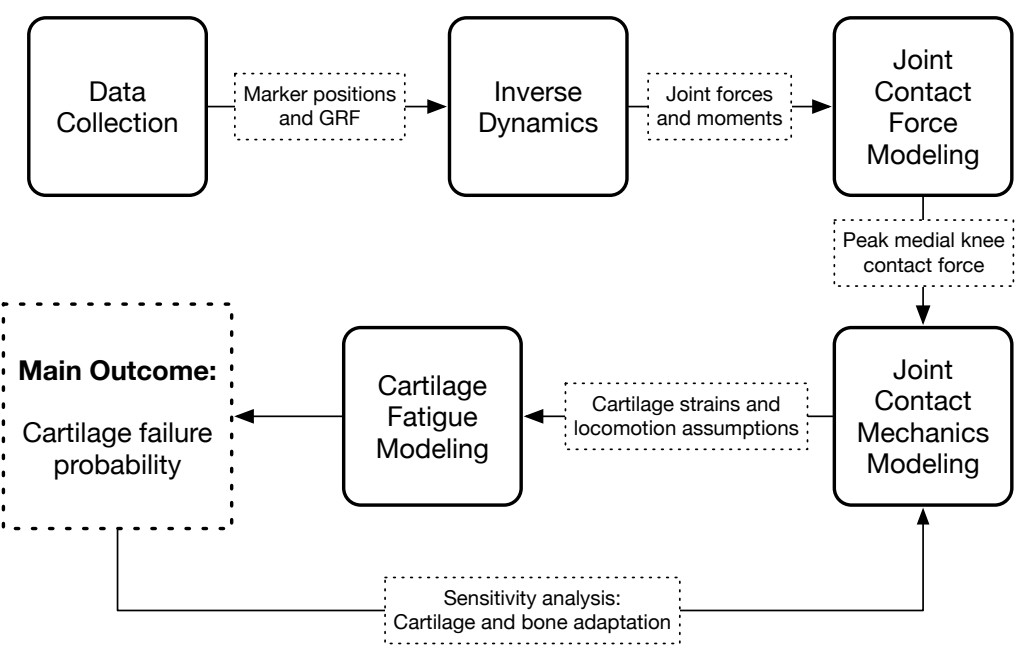

**Figure 1** **Study workflow.** Workflow of the study. Arrows going in are input data and arrows coming out are calculated output variables.

Therefore, the purpose of this study was to combine musculoskeletal modeling with probabilistic fatigue modeling to compare (i) peak medial knee cartilage strains in walking and running, (ii) lifetime failure probability of medial knee cartilage when traveling a given daily distance with and without running, and (iii) the probability of the knee cartilage withstanding a lifetime of running with and without adaptation. While there are many other relevant scales for examining structural fatigue in osteoarthritis, e.g., whole-joint, cellular, molecular, here we focused on the tissue level due to the simplicity of the applicable models, the availability of tissue-level fatigue testing data, and the ease of comparison to typical clinical definitions of "structural" osteoarthritis.

## METHODS

The study consisted of an experimental gait analysis of 22 participants, followed by four stages of computer modeling that used the gait data to estimate long-term medialknee cartilage failure probability: inverse dynamics, joint contact force modeling, joint contact mechanics modeling, and cartilage fatigue modeling. The latter two modeling stages were repeated several times in a sensitivity analysis. The workflow of the study is diagrammed in Fig. 1.

### Participants

Participants were 22 healthy young adults with (mean ± SD) age 23 ± 3 years, height 1.73 ± 0.08 m, and mass 68.9 ± 11.7 kg. Participants gave written informed consent prior to participating. All protocols were approved by the University of Maryland's institutional

review board (IRBnet ID 671084). Participants were included if they reported no injuries or conditions to the lower limb or lower back that affected their ability to walk or exercise in the past year. The self-reported participant sex distribution was 12 men and 10 women. The self-reported participant race distribution was 14 Caucasian, two African American, six Other.

## Gait analysis

Participants walked and ran at self-selected "normal and comfortable" speeds around an indoor 50-m track wearing their own usual running shoes. A straight section of the track approximately 12 m in length was instrumented with 12 motion capture cameras (Vicon, 200 Hz) and eight force platforms (Kistler, 1,000 Hz). The cameras measured positions of 40 retroreflective markers on the lower limbs and pelvis (*Krupenevich, Pruziner & Miller, 2017*). Markers on the medial toe, ankle, and knee joints were present only during a standing calibration trial and were removed during walking/running trials. Subjects walked (or ran) around the track for five minutes while force and motion data were sampled. The order of walking and running conditions was balanced between participants.

Data from all complete strides were proceeded with inverse dynamics analysis in Visual3D software (C-Motion) to compute joint angular positions and resultant kinetics using six-degree-of-freedom segment models defined from the standing calibration trial data. The knee joint center was defined as the midpoint of the femoral epicondyle markers and the ankle joint center as the midpoint of the malleoli markers. These joints centers were then reconstructed as "virtual" markers during the walking/running trials based on their positions relative to other markers in the standing trial. The long axis of the shank was the unit vector between the knee and ankle joint centers. The frontal axis of the shank was the cross product of the long axis and the vector between the femoral epicondyle markers. The lateral axis of the shank was the cross product of the frontal and longitudinal axes. The inverse dynamics joint kinetics were expressed in this shank reference frame.

## Knee load modeling

Medial tibiofemoral contact forces were calculated from the lower limb joint angles and kinetics using a reduction modeling approach (*DeVita & Hortobagyi, 2001*). The patellar tendon force was determined by dividing the knee extension moment by the patellar tendon moment arm, after subtracting the moments due to antagonist forces by the hamstrings (determined from the hip extension moment) and gastrocnemius (determined from the ankle plantarflexion moment). Forces in the anterior cruciate, posterior cruciate, lateral collateral, and medial collateral ligaments were calculated according to *Morrison (1968)*. Moment arms were defined as quadratic functions of joint angles, using data from *Menegaldo, Fleury & Weber (2004)* except for the patellar tendon, which were from *Herzog & Read (1993)*. Achilles and patellar tendon moment arms could increase by up to 1.0 cm as linear functions of the tendon force (*Maganaris, Baltzopoulos & Sargeant, 1998*; *Tsaopoulos et al., 2007*). Muscle physiological cross-sectional areas were defined as average values from *Miller (2018)* and were used as the weights by which the joint moments were distributed into agonist muscles. Maximum isometric forces were calculated by multiplying

the physiological cross-sectional areas by an assumed specific tension of 40 N/cm $^2$. This value has a small effect on the moment arm sensitivity to muscle forces note above but does not affect study outcomes over the typical range of 20–60 N/cm$^2$.

Medial knee contact forces were then calculated by balancing internal and external moments of force about the lateral tibiofemoral contact point (*Miller, Esterson & Shim, 2015*). This approach produces estimates of knee contact forces and medial-lateral force distributions that compare reasonably well with measurements from instrumented joint replacements, with typical errors in the range of 0.3–0.9x bodyweight (BW) (*Miller, Esterson & Shim, 2015*; *Dumas et al., 2019*).

## Knee contact mechanics

The peak knee contact forces during the stance phase determined from an average of at least five strides per subject and were input to a model of medial knee contact mechanics to determine the strain distribution on the tibiofemoral cartilage. The contact model was based on *Nuño & Ahmed (2001)* and is summarized visually in Fig. 2, drawn using the actual dimensions of the model in the mid-sagittal plane. The medial femoral condyle was modeled as two circular arcs representing its anterior and posterior aspects in the sagittal plane, and as a single arc in the frontal plane. The tibial plateau was a concave bowl. The bone geometric parameters were the mean values from 12 cadaver knees referenced in *Nuño & Ahmed (2001)* and the same values were used for all participants.

The tibia was fixed in space and the femur had two degrees of freedom: the height of the flexion axis, and the flexion angle. The tibiofemoral cartilage was modeled as an array of contact elements on the tibial plateau, separated by distance $d = 0.5$ mm (element area $d^2 = 0.25$ mm$^2$) and with unloaded length $h = 5.0$ mm. Each element had a nonlinear elastic stress–strain relationship to account for the large deformations in running (*Blankevoort et al., 1991*):

$$\sigma_i = -E_i \bullet \ln(1 - \epsilon_i) \tag{1}$$

$$\epsilon_i = y_i / h \tag{2}$$

and the knee contact force was:

$$F = d^2 \sum_{i=1}^{N} \sigma_i \tag{3}$$

where $\sigma_i$ is the contact stress of element $i$, $E_i$ is the cartilage/meniscus modulus, $y_i$ is the spring compression, and $N$ is the number of contact elements (7,326 with $d = 0.5$ mm). The modulus varied with element position because some elements were covered by the medial meniscus. The flexion angle was set to the angle from the gait analysis data at which the peak medial knee contact force occurred, and the flexion axis height was calculated such that the contact force equaled the participant's average value from the gait analysis data. With the different radii of curvature for the anterior and posterior femoral aspects, the knee flexion angle affected the area of cartilage loaded, as is typically observed in cadavers (*Henderson, Higginson & Barrance, 2011*). We did not model translational kinematics of the knee because the medial femoral condyle appears to remain near the center of the
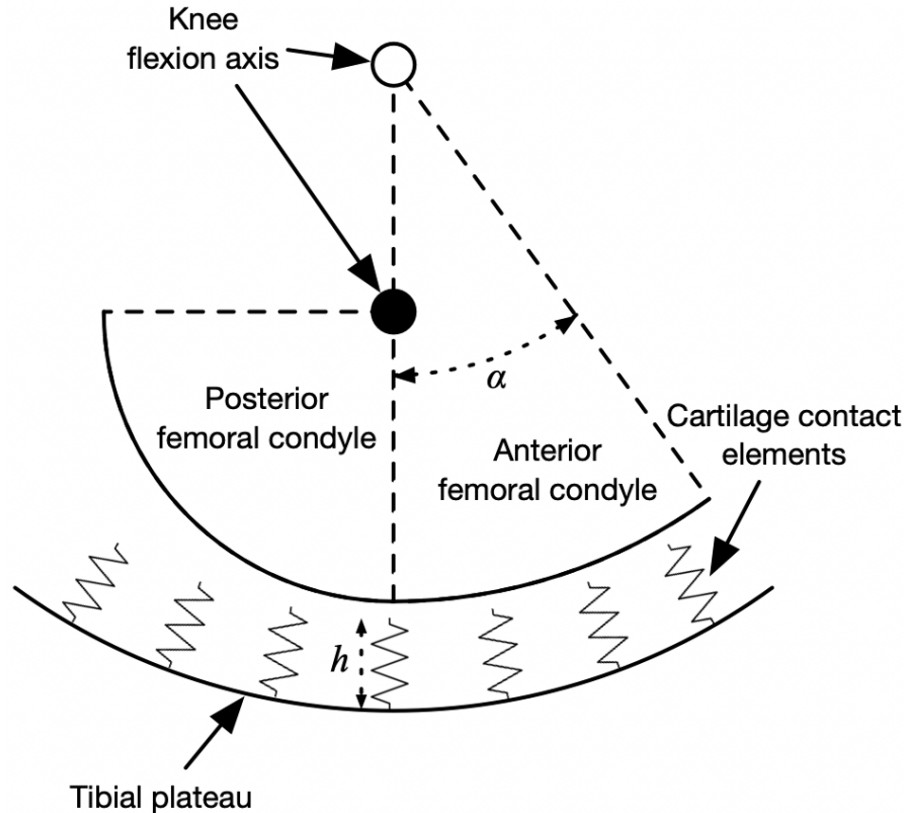

**Figure 2** **Diagram of knee contact mechanics model in the sagittal plane.** Diagram of the medial knee contact mechanics model in the sagittal plane (*Nuño & Ahmed, 2001*). The tibia is fixed in space and the femur has two degrees of freedom: the height of the flexion axis relative to the tibia, and the angle of the flexion axis relative to the tibia. The location of the flexion axis within the femur switches from the empty circle to the filled circle when the flexion angle exceeds the angle $\alpha$. Compression of the contact element springs from the unloaded length produces stresses, strains, and contact forces. The actual model had thousands of contact elements; only a few are shown here for visualization.

tibial plateau with flexion (*DeFrate et al., 2004*). Bone was assumed to be rigid. The moduli for the femoral cartilage, uncovered tibial cartilage, and covered tibial cartilage, were 8.6, 4.0, and 10.1 MPa, respectively (*Shepherd & Seedhom, 1999*). The modulus of the medial meniscus was 1.3 MPa and the meniscus covered the outer 46% of the tibial plateau (*Danso et al., 2015*; *Bloecker et al., 2013*). The coverage of the meniscus is visualized in Fig. 3. The cartilage and meniscus were assumed to be near-incompressible, with Poisson's ratio $v = 0.45$. The unloaded cartilage thickness (tibial + femoral) was 5.0 mm (*Liu et al., 2010*). There were no geometric differences between model regions covered vs. uncovered by meniscus; the only difference was the modulus for contact springs in these regions.

Concerning the elastic moduli, the apparent compressive stiffness of cartilage depends on loading rate, e.g., the "equilibrium" vs. "dynamic" modulus of cartilage can differ by a factor of $\sim$10x (*McCutchen, 1962*). However, for a range of loading rates relevant to human locomotion, the dynamic modulus of cartilage varies by $\sim$1.2x at most (*Oloyede, Flachsmann & Broom, 1992*). Cartilage strains in running were therefore calculated both

**Figure 3** **Axial view of medial knee model.** Bone (white), cartilage (black), and meniscus (gray) for the medial knee, looking down through the long axis of the tibia. The image on the left is the Open Knee finite element model (*Erdemir, 2016*), compared to the present discrete-element model on the right.

with and without multiplying $E_i$ in Eq. (1) by a factor of 1.2. Which strains were used depended in the hypothesis being tested. The present model is unable to predict how material properties like modulus respond to changes in applied load parameters like loading rate, as it lacks explicit representations of elements like interstitial fluid flow that affect these mechanics in cartilage; this is a limitation of using a model of minimum complexity for addressing this research question.

## Cartilage failure probability

"Failure" of the medial tibiofemoral cartilage was defined as macroscopic plastic deformation, similar to the superficial fibrillation seen in early-stage "structural" osteoarthritis (*Weightman, Freeman & Swanson, 1973*). For each cartilage element in the contact mechanics model, the probability of failure over an adult lifespan of 60 years was calculated using Taylor's phenomenological model of tissue damage, repair, and adaptation (*Taylor, 1998*; *Taylor & Kuiper, 2001*; *Taylor, Casolari & Bignardi, 2004*).

### *Damage*

The main equation to the model is the cumulative probability of failure $P_f$ at time $t$ for a tissue that sustains $\dot{D}/L$ loading cycles per unit time, where $\dot{D}$ is the distance traveled per unit time, e.g., meters per day, and $L$ is the stride length. Each cycle has peak stress $\sigma$ and strain $\epsilon$. The failure probability equation has the form of a Weibull cumulative probability

function:

$$P_f(t) = 1 - \exp\left[-\left(\frac{V}{V_{ref}}\right)\left(\frac{t}{t_f}\right)^{m/n}\right]. \tag{4}$$

The relationship between the time until failure $t_f$ and element strain $\epsilon_i$ (the "fatigue life" or "strain-life" relationship) is assumed to be a power law:

$$t_f = \left(\frac{CL}{\dot{D}}\right)(b\epsilon_i)^{-n} \tag{5}$$

where $m$, $C$, $n$, and $b$ are constants, $V$ is the volume of material stressed by $\sigma_i$, and $V_{ref}$ is the referenced stressed volume in the data from which the other parameters are determined. The damage model is formulated with strain, rather than stress, load, adduction moment, etc. as its key input variable because strain is the closest analogue for actual structural "damage" in the form of plastic deformation.

Values for $C$ and $n$ are determined by fitting a power law for cycles rather than time ($N_f = C\epsilon^{-n}$, where $N_f$ is the cycles until failure) to data from *Riemenschneider et al. (2019)*. The fit to these data is presented in Fig. 4. Note that the triangles in Fig. 4 were "runout" samples from *Riemenschneider et al. (2019)* that were undamaged at the testing limit of 100,000 loading cycles and were **not** included in the fitting process. The runout samples with the lower strain levels may represent the hypothetical "endurance limit" of the cartilage, where a "lifetime" of loading cycles can be sustained before failure. At a strain of 0.23, roughly the strain of walking (*Liu et al., 2010*), the model predicts failure at $\sim$12 billion cycles, or 64 years of walking 10,000 steps/day. The values for $m$ and $b$ are determined by:

i. Using the power law to compute the strain resulting in failure at a number of loading cycles relevant to a human lifespan ($1 \bullet 10^7$ was used)
ii. Assuming the standard deviation of scatter in failure strains around this strain is the same as the scatter at other numbers of loading cycles in the Riemenschneider data, $\sim$0.025 strains
iii. Drawing random estimated failure strains at $1 \bullet 10^7$ loading cycles from the standard normal distribution defined by this mean and standard deviation
iv. Fitting Eq. (4) to the data from (iii)

The fit to these data is presented in Fig. 4. Table 1 presents the parameter values for the damage model.

The meaning of the parameters in Eqs. (4) and (5) are as follows. The time until failure $t_f$ is the distribution's scale parameter, the time at which 63.2% of cases would be expected to fail when loaded with strain $b\epsilon_i$ for $\dot{D}/L$ cycles per day. The parameter $m$ is the distribution's shape parameter and indicates the scatter in the experimental failure data, with larger values of $m$ for data that fail over a narrower range of stress levels with a given number of loading cycles. The parameter $n$ is the slope of the fatigue-life relationship on a log–log plot. The remaining parameters $b$ and $C$ are simply curve-fitting constants, assuming the strain that

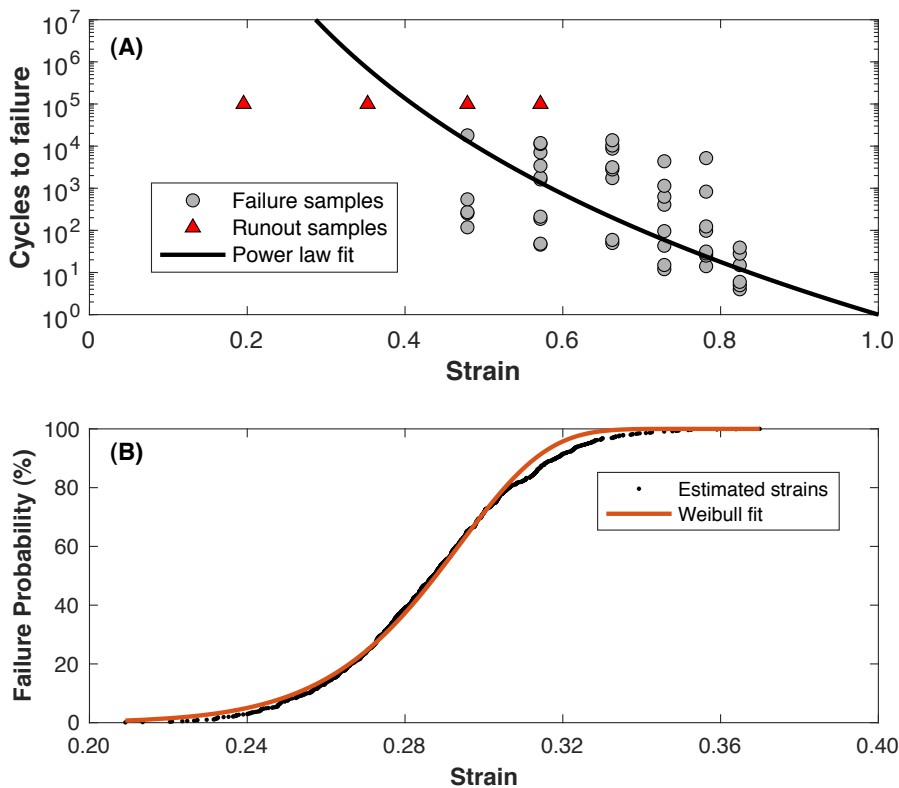

**Figure 4** **Fatigue life model.** (A) Strain vs. cycles until failure from the referenced data (*Riemenschneider et al., 2019*) and the power-law model fit (solid line). Triangle symbols are runout samples that survived 100,000 cycles of the indicated strain without failure; these samples were not included in the model fitting. (B) Estimated failure strain distribution for $1 \cdot 10^7$ loading cycles (symbols), fit with a Weibull cumulative distribution function (line).

**Table 1** **Failure probability model parameters.**

| Symbol | Description | Value |
|--------|-------------|-------|
| $C$ | Power law coefficient | 1.0 |
| $n$ | Power law exponent | 12.9 |
| $b$ | Weibull coefficient | 1.03 |
| $m$ | Weibull exponent | 14.3 |
| $V_{ref}$ | Referenced stressed volume | 78.5 mm$^3$ |
| $\dot{D}$ | Distance traveled per day | 6,000 m/day |
| $t_r$ | Cartilage repair time constant | 5.0 years |
| $v$ | Cartilage repair exponent | 5.2 |

causes failure with 63.2% probability is a constant offset from the input strain $\epsilon_i$. The damage model does not explicitly include degradation of cartilage structure and material properties as functions of cumulative damage. These degradations are included implicitly in the *Riemenschneider et al. (2019)* fatigue-life data the model is fit to.

### Repair

Equation (4) is the probability of failure for materials with no ability to repair damage, such as a cartilage explant removed from a living system. Living cartilage normally has no connections to the nervous and circulatory systems and is thus considered to have little ability to naturally repair damage that has not reached the underlying bone, suggesting Eq. (4) should not be augmented to include repair. However, there is evidence that living cartilage in young adults possesses some natural ability to at least partially recover from such damage over time periods of several years (*Nakamura et al., 2008*). The probability of repair is included in the failure model as a second Weibull function:

$$P_r = 1 - \exp\left[-\left(\frac{t}{t_r}\right)^v\right] \tag{6}$$

where $t_r$ and $v$ are again the scale and shape parameters: $t_r$ is the time after which repair would be expected in 63.2% of cases of damage, and $v$ depends on the scatter of repair times between subjects and effects the rate at which $P_r$ increases with increasing $t$.

Repair is included in Eq. (4) by first deriving the probability density function $Q_f = dP_f/dt$, which gives the instantaneous probability of failure at a given time:

$$Q_f = \left(\frac{Vm}{nV_{ref}t_f}\right)\left(\frac{t}{t_f}\right)^{m/n-1}\exp\left[-\left(\frac{V}{V_{ref}}\right)\left(\frac{t}{t_f}\right)^{m/n}\right] \tag{7}$$

The instantaneous probability of failure is multiplied by the cumulative probability that repair has *not* occurred yet $(1-P_r)$, then this product is integrated over time to determine the probability of failure with repair, $P_{fr}$:

$$P_{fr} = \int_0^t \left[Q_f \bullet (1-P_r)\right] dt. \tag{8}$$

This model of repair is rather optimistic for cartilage, as it assumes that most cases of damage occurring at time $t$ will fully repair by time $t + t_r$, and that nearly all cases of damage will fully repair eventually. A generous estimate of repair capacity avoids biasing the model in favor of our "cartilage conditioning" hypothesis (*Miller, 2017*).

### Adaptation

In a healthy state, mechanical loading from physical activity in theory can stimulate a metabolic response in chondrocytes to strengthen the extracellular matrix (*Seedhom, 2006*; *Andriacchi, Koo & Scanlan, 2009*). The turnover of collagen in human knee cartilage appears to cease upon reaching skeletal maturity, age $\sim$ 20 years, suggesting the ability of collagen to adapt is limited in adulthood (*Heinemeier et al., 2016*), but adaptations to other elements of cartilage regulated by chondrocyte metabolism are still feasible. For example, *Van Ginckel et al. (2010)* reported that 10 weeks of running in untrained females increased knee cartilage dGEMRIC index, an estimate of the glycosaminoglycan content that affects cartilage stiffness (*Samosky et al., 2005*; *Baldassarri et al., 2007*).

Adaptation was included in the model by recomputing the cartilage element stresses and strains with different knee model parameters reflecting positive remodeling, then recomputing failure probabilities with the new strains. Data on human running concerning

this remodeling is scarce aside from brief time periods of training with indirect measures of material properties (e.g., *Van Ginckel et al., 2010*), but animal models suggest moderate running can increase the cartilage compressive modulus, cartilage thickness, and/or joint congruency via bone geometry (*Jurvelin et al., 1986*; *Kiviranta et al., 1988*; *Oettmeier et al., 1992*; *Firth & Rogers, 2005*; *Hamann et al., 2014*). The parameters adjusted were therefore the cartilage elastic modulus $E$, cartilage unloaded length $h$, and the tibiofemoral radii of curvature.

## Implementation & statistics

The first hypothesis (contact mechanics cause peak strains to be similar between walking and running) was tested by comparing the greatest cartilage strains in walking vs. running. The peak medial knee contact forces were input to the contact mechanics model. From the resulting spatial distributions of strains on the cartilage contact elements, the greatest strain in any element for each determined and compared statistically between walking and running. A matched-pair Student's $t$-test was first performed. If the null hypothesis of equal strains could not be rejected ($p_{NHST} > 0.05$), two one-sided tests of equivalence were performed (*Lakens, 2017*). The equivalence bounds were set to effect sizes of $\pm 0.05$ and the equivalence test was accepted if $p_{TOST} < 0.05$. This was comparison between walking and running was performed twice, once with and once without the cartilage modulus adjusted for running. These same analyses were performed on two other related variables of interest: the peak medial knee contact force, and medial knee cumulative load per unit distance traveled (average contact force divided by stride length; *Miller et al., 2014*).

The second hypothesis (cumulative damage over a given distance is similar between walking and running) was tested by calculating failure probabilities over an assumed adult lifespan of 60 years (age ~23–83 years for these subjects), assuming a daily distance traveled of 6 km. Given the step lengths measured in this study, this distance equates to an average of 7,756 steps per day, about one standard error above the mean reported by *Tudor-Locke, Johnson & Katzmarzyk (2009)* for healthy American adults when excluding steps taken at an "inactive" intensity level. The comparison was made between failure probabilities without repair or adaptation when walking all 6 km/day vs. walking 3 km/day and running 3 km/day. The same comparison was also made between failure probabilities that included repair. The statistical approach was the same tests used on the first hypothesis.

The third hypothesis (strains in running condition cartilage such that the risk of developing osteoarthritis does not increase) was tested by increasing the adaptation parameters (cartilage elastic modulus, cartilage unloaded thickness, tibiofemoral radii of curvature) by increments of +0.05 times the between-subjects standard deviations from their source data (Table 2) and recomputing the failure probabilities with the new parameter value(s). This sensitivity analysis was performed on each parameter in isolation, holding the other two at their original values, and lastly with all three parameters increased simultaneously. Changes in parameter values were performed in a single step, before calculating the failure probability. In other words, the parameter values did not gradually change during the simulated 60-year time period. The assumption here is that the adaptation had already occurred from the participant's lifestyle/exercise prior to this

**Table 2  Knee contact mechanics model parameter values.** Default values were the referenced means and adapted values were determined using the referenced standard deviations (SD).

| Parameter | Reference | Mean | SD | Range |
|---|---|---|---|---|
| Femoral cartilage modulus (MPa) | *Shepherd & Seedhom (1999)* | 8.6 | 2.3 | 4.3–13.0 |
| Covered tibial cartilage modulus (MPa) | *Shepherd & Seedhom (1999)* | 10.1 | 2.2 | 5.9–13.6 |
| Uncovered tibial cartilage modulus (MPa) | *Shepherd & Seedhom (1999)* | 4.0 | 1.6 | 2.8–7.8 |
| Meniscus modulus (MPa) | *Danso et al. (2015)* | 1.3 | 0.6 | [a]0.1–2.5 |
| Unloaded cartilage thickness (mm) | *Liu et al. (2010)* | 5.0 | 1.5 | [a]2.0–8.0 |
| Sagittal anterior femur radius (mm) | *Nuño & Ahmed (2001)* | 35.0 | 4.1 | 27.6–41.8 |
| Frontal tibia radius (mm) | *Nuño & Ahmed (2001)* | 21.0 | 2.3 | 15.9–23.6 |

**Notes.**
[a]Ranges for meniscus modulus and unloaded cartilage thickness were estimate as four times the standard deviation, centered on the mean, because the referenced studies did not report ranges.

period and is motivated by evidence that the adaptive capacity of cartilage is limited beyond skeletal maturity, age ∼20-25 years (*Heinemeier et al., 2016*). From this analysis, we determined the increases in parameter values needed for the mean failure probability of walking 3 km/day and running 3 km/day to equal the mean failure probability of walking 6 km/day (no running) with the original unadjusted parameter values.

For both the second and third hypotheses, all calculations were performed using the strains estimated by assuming the cartilage modulus is 1.2x greater in running vs. walking. These strains were smaller than the strains estimated by assuming equal moduli and incur less damage. Similar to the generous assumptions in the repair model, a factor of 1.2x is on the upper end of plausible differences in the compressive modulus due to loading rate differences in walking and running (*Oloyede, Flachsmann & Broom, 1992*). These assumptions were used here to avoid favoring the expected result that substantial adaptation would be necessary to reduce the cartilage failure probability when running to "unremarkable" levels (*Miller, 2017*).

## RESULTS

The average walking speed was 1.52 ± 0.18 m/s, with average stride length 1.55 ± 0.15 m. The average running speed was 2.58 ± 0.20 m/s, with average stride length 1.95 ± 0.18 m. The validity of various sub-model components was examined by comparisons to *in vivo* data in the literature. The medial knee loads are compared in Fig. 5 to instrumented knee replacement data from subject "K8L" in the OrthoLoad database, an older adult male with bodyweight 755 N (*Bergmann et al., 2014*). This subject was chosen for comparison because they were the only subject in the database tested at a walking speed similar to the present subjects. K8L walked slightly slower on average than the present subjects (1.39 vs. 1.52 ± 0.18 m/s), and their peak medial knee loads were about one standard deviation below the mean of the model-based estimates for the present participants (2.42 vs. 2.90 ± 0.55 BW). The medial knee load at heel-strike was greater in K8L's measurements than in the model estimates (0.98 vs. 0.15 ± 0.09 BW), although instrumented knee data may be of limited accuracy below ∼1000 N (*Halder et al., 2012*). K8L's peak lateral contact force was also lower than the average model estimate for these participants (0.65 vs. 1.12
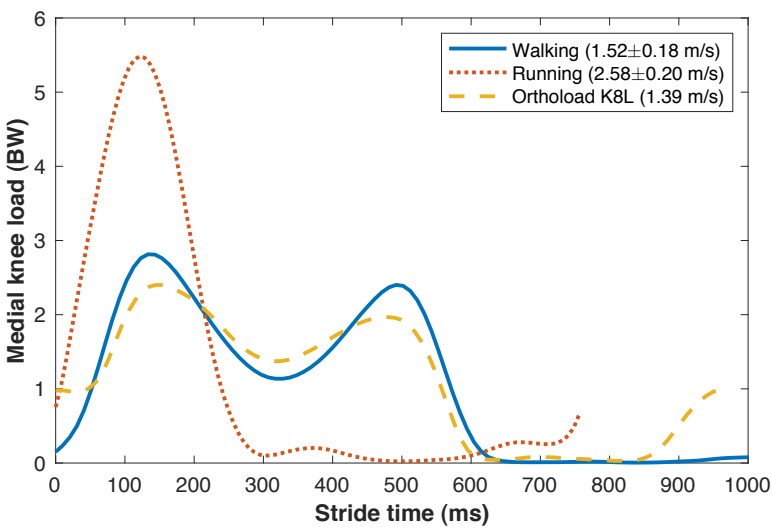

**Figure 5  Medial knee contact forces.** Average medial knee contact forces during the strides of walking and running at self-selected "normal and comfortable" speeds, in multiples of bodyweight (BW). The plots begin and end at consecutive ipsilateral heelstrikes. Orthoload data are subject K8L from *Bergmann et al. (2014)* walking at their fastest measured speed.

$\pm$ 0.28 BW) but within the range of estimates (0.54–1.71 BW). There are no instrumented knee data at comparable running speeds, but the peak medial knee loads estimated here for running (5.59 BW @ 2.58 m/s) were slightly smaller than those in another study with participants running at a slightly faster speed (*Willy et al., 2016*): 6.14 BW @ 3.18 m/s) The greatest compressive strains on the medial knee cartilage in walking (23.0 $\pm$ 4.4%) were similar to those reported by *Liu et al. (2010)* from fluoroscopy measurements of eight subjects during treadmill walking (23 $\pm$ 6%). For the case of walking 6 km/day, the lifetime failure probability of the medial knee cartilage was 44.2 $\pm$ 42.4% without repair, and 13.4 $\pm$ 21.7% with repair. In epidemiology studies, *Murphy et al. (2008)* estimated a lifetime risk of incident knee osteoarthritis in American adults of 32.2–52.5%, and *Losina et al. (2013)* estimated 13.8%.

The peak medial knee load was greater in running vs. walking (Fig. 5: 5.59 $\pm$ 0.99 vs. 2.90 $\pm$ 0.55 BW, $p_{NHST} = 3.28 \bullet 10^{-13}$). The greatest cartilage strains occurring from this load were also greater in running vs. walking (Fig. 6), both when assuming the same modulus for both gaits (44.7 $\pm$ 8.2 vs. 23.0 $\pm$ 4.4%, $p_{NHST} = 3.43 \bullet 10^{-14}$) and when assuming the modulus was 1.2x greater in running due to the greater loading rate (39.7 $\pm$ 7.5 vs. 23.0 $\pm$ 4.4%, $p_{NHST} = 8.03 \bullet 10^{-13}$). Cumulative load on the medial knee did not differ between running vs. walking (0.64 $\pm$ 0.09 vs. 0.64 $\pm$ 0.11 BW/m, $p_{NHST} =0.71$, $p_{TOST}=0.0017$).

The time to estimated medial tibiofemoral cartilage failure depended on how much walking and running was undertaken and on the inclusion of repair (Fig. 7). Without repair, the lifetime failure probability of the medial knee cartilage was greater when walking and running (3 km/day each) vs. only walking (99.0 $\pm$ 4.6 vs. 44.2 $\pm$ 42.4%, $p_{NHST} = 3.96 \bullet 10^{-6}$). Including repair had only a small effect on the failure probability

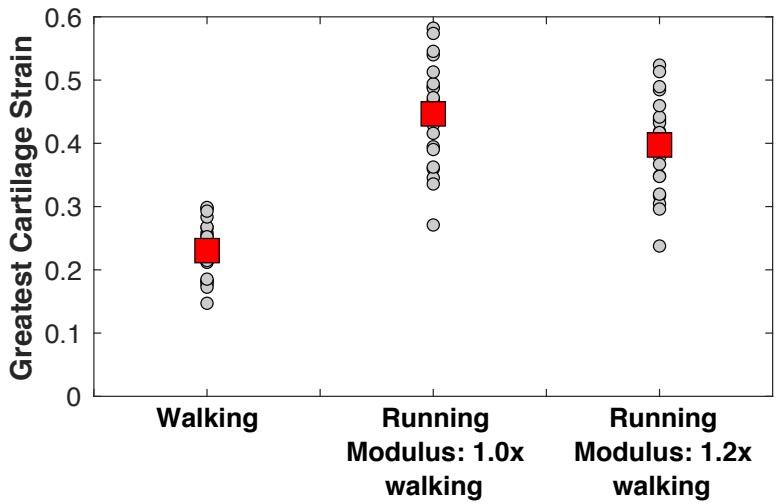

**Figure 6** **Peak cartilage strains.** Greatest medial tibiofemoral cartilage stains occurring from the peak medial knee contact force in walking and running at a self-selected speeds. Smaller circles are individual subjects. Larger squares are means. Running strains are shown with and without the cartilage modulus stiffened relative to the modulus for walking.

when running, and the lifetime failure probability was still greater when walking and running vs. only walking (95.2 ± 19.5 vs. 13.4 ± 21.7%, $p_{NHST} = 2.59 \bullet 10^{-12}$).

Reducing the mean lifetime failure probability when walking and running to just under the mean when only walking (threshold 13.4%) was only possible when including adaptation in the model (Fig. 7). Changing any one of the cartilage elastic modulus, cartilage unloaded thickness, or tibiofemoral radii of curvature required rather large, seemingly implausible increases to reach this threshold: +2.60 SD increase in cartilage elastic modulus (e.g., 10.1→15.8 MPa for uncovered tibiofemoral cartilage), +5.35 SD increase in cartilage unloaded thickness (5.0→13.0 mm), and +6.70 SD increase in tibiofemoral radii of curvature (e.g., 35.0→62.5 mm for the anterior aspect of the femoral condyle). When all three parameters were adjusted simultaneously, the threshold failure probability was reached with an increase of +1.15 SD to each parameter (10.1→12.6 MPa, 5.0→6.7 mm, 35.0→39.7 mm).

When examining the estimated failure probabilities at age 55 (Fig. 8), a common age of knee osteoarthritis diagnosis (*Losina et al., 2013*), with no running and with cartilage repair modeled, only 4/22 subjects had failure probabilities above 20%. With 3 km/day of running and no adaptation, 17/22 subjects had a failure probability of at least 89% even with repair, and 21/22 subjects had a failure probability of at least 27% (the one exceptional subject had the lowest peak joint loads in the sample). With +1.15 SD increases to the three adaptation parameters, only four subjects had failure probabilities above 17% when running 3 km/day.

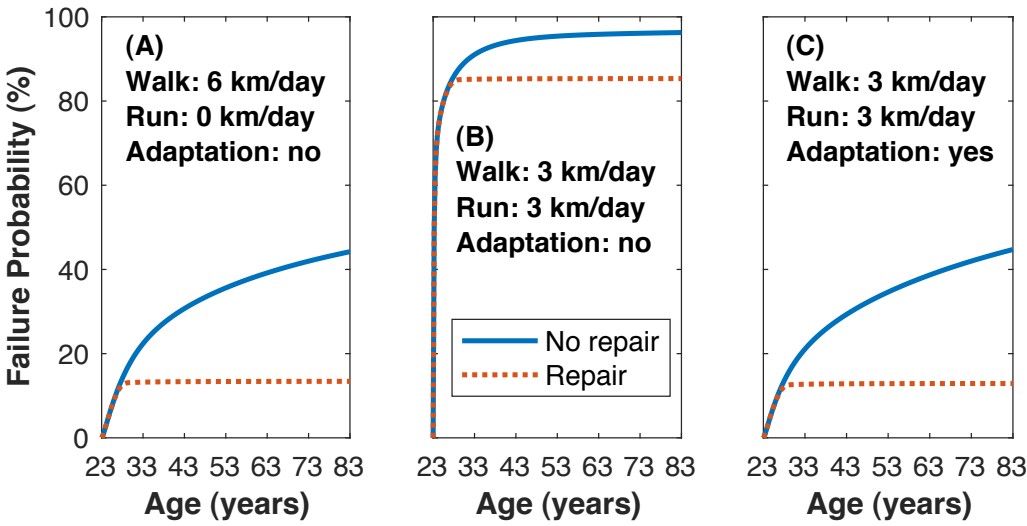

**Figure 7  Failure probability time series.** Average medial tibiofemoral cartilage failure probabilities over 60 years of adulthood when traveling 6 km/day. (A) Walking 6 km/day, no running. (B) Walking 3 km/-day and running 3 km/day, no cartilage adaptation from running. (C) Walking 3 km/day and running 3 km/day, cartilage adaptation from running. The probabilities with adaptation were computed with +1.15 standard deviations increase to all three identified adaptation parameters: cartilage modulus, cartilage thickness, tibiofemoral congruence.

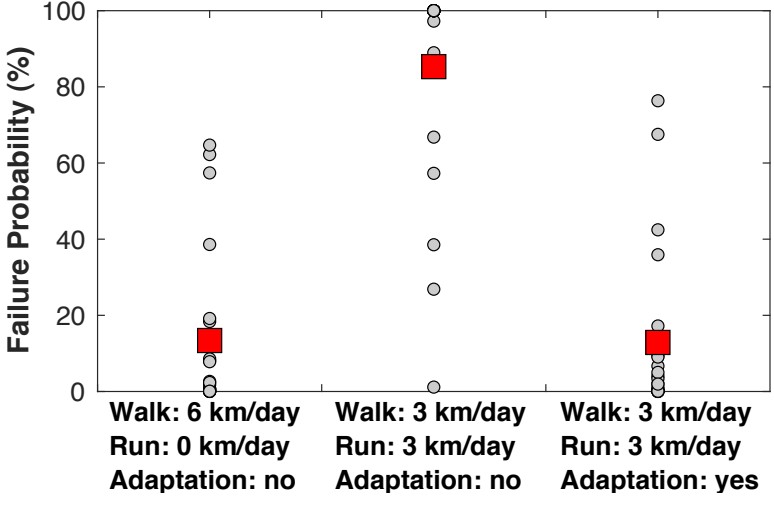

**Figure 8  Failure probabilities at age 55.** Medial tibiofemoral cartilage failure probabilities at age 55 when traveling 6 km/day. *Left:* walking 6 km/day, no running. *Middle:* walking 3 km/day and running 3 km/day, no cartilage adaptation from running. *Right:* walking 3 km/day and running 3 km/day, cartilage adaptation from running. The probabilities with adaptation were computed with +1.15 standard deviations increases to all three adaptation parameters: cartilage modulus, cartilage thickness, and tibiofemoral congruence. All probabilities included repair. Smaller circles are individual subjects. Larger squares are means.

## DISCUSSION

The purpose of this study was to examine three hypotheses for explaining why long-distance running is generally not associated with an increased risk of knee osteoarthritis. Given the challenges of studying osteoarthritis across the lifespan, a model-based evaluation of joint loading, cumulative damage, and cartilage failure probability was performed. The first hypothesis was that contact mechanics between the femoral condyle and tibial plateau result in cartilage strains in running that are similar to the strains of walking. This hypothesis is rejected, based on Fig. 6: the greatest cartilage strain was on average about twice as large in running vs. walking, even when making a generous assumption that cartilage is 20% stiffer in running due to faster loading rates. Materials testing suggests the loading rates of walking and running, which differed here by a factor of 1.8x on average, are both within the "high stiffness" regime of cartilage where changes in loading rate over an order of magnitude do not appreciably affect the dynamic compressive modulus (*Oloyede, Flachsmann & Broom, 1992*). We caution, however, that the speeds in this study were fairly fast on average for walking (1.52 m/s) and fairly slow for running (2.58 m/s). Gaits with greater differences in speed could have greater sensitivity of material properties to loading conditions that would need to be included in a model of cartilage mechanics. The present model is unable to simulate this sensitivity in a "predictive" way; material properties must be specified ahead of time for each condition.

The second hypothesis was that the rate of damage accumulation per unit distance traveled would be similar in walking and running. This hypothesis is also rejected, based on Fig. 8: even though the cumulative load per unit distance was similar between walking and running, replacing 3 km/day of walking with 3 km/day of running increased the probability of cartilage failure. The third hypothesis was that mechanical loading from running conditions cartilage to safely sustain these loads, such that the risk of knee osteoarthritis does not appreciably increase. This hypothesis is accepted, based on Fig. 7: with modest increases in three plausibly adaptable parameters (cartilage modulus, cartilage thickness, bone geometry), it was possible to replace 3 km/day of walking with 3 km/day of running without increasing the probability of cartilage failure.

The main limitation that could temper these conclusions is that the approach involves a great deal of model-based estimation. It is possible these results are unique to this particular sequence of models and that other models with different levels of complexity would produce different results. We make no claims of these results being definitive or of direct clinical relevance, e.g., how much running can be tolerated without substantial osteoarthritis risk. Ideally these research questions would be investigated in long-term experiments on human subjects, but such experiments are impractical due to the long timecourse over which osteoarthritis initiates, and the difficulty of directly quantifying and manipulating cartilage material properties in living subjects. Conceptually similar studies on human subjects have been performed over time periods of 2–3 months, with indirect measures of cartilage material properties that support the cartilage conditioning hypothesis (*Van Ginckel et al., 2010*; *Celik et al., 2013*). The modeling approach used here was developed by *Taylor, Casolari & Bignardi (2004)* for studying bone stress injuries. Its accuracy in predicting bone
stress injury incidence has not been directly assessed, but it produces injury probabilities similar to incidence rates of bone stress injuries in runners (*Taylor, Casolari & Bignardi, 2004*; *Edwards et al., 2009*). The present implementation of the model produced medial knee cartilage failure probabilities similar to estimates of lifetime incidence rates of medial knee osteoarthritis in non-running adults (*Murphy et al., 2008*; *Losina et al., 2013*) from estimates of peak joint loads and compressive strains similar to *in vivo* measurements in walking adults (*Bergmann et al., 2014*; *Liu et al., 2010*). Therefore, while this modeling approach remains unvalidated for the purpose of predicting subject-specific development of knee osteoarthritis, it appears to produce reasonable estimates on average of joint loads, cartilage strains, and structural failure probabilities.

Concerning the fatigue-life model specifically (Eq. (5), Fig. 3), performing this kind of modeling is challenging with the paucity of data on cartilage fatigue in the literature. *Riemenschneider et al. (2019)*, published just last year, is the only published data on fatigue of cartilage in any animal in response to compressive loading. Several other studies examined fatigue of human hip and knee cartilage, but loaded strips of cartilage in tension along the collagen fibers (*Weightman, Freeman & Swanson, 1973*; *Weightman, Chappell & Jenkins, 1978*; *Bellucci & Seedhom, 2001*), which would require a much more complex model of cartilage mechanics to transform gait analysis data into collagen fiber strains. The present model was attractive for its simplicity, requiring only an estimate of the macroscopic strain distribution of cartilage as a whole material, but this simplicity prevents inferences on loading and failure of any particular elements of cartilage, e.g., collagen fibers, chondrocytes. Relatedly, the simplicity and phenomenological nature of the present modeling prevents conclusions on any specific smaller-scale cartilage mechanics and mechanisms of failure that were not modeled here, e.g., interstitial fluid flow (*Barker & Seedhom, 2001*) and delamination (*Durney et al., 2020*). Other models of cartilage mechanics and damage with more mechanistic underpinnings may be better suited to those tasks (e.g., *Hosseini et al., 2014*; *Gardiner et al., 2016*; *Mononen et al., 2018*). The specifics of these models and how they differ from the present model is beyond the scope of this study.

Another limitation is that the modeling of joint loads, contact mechanics, and cartilage failure used generic model parameters. The severity of this limitation is reduced by the repeated-measures design of the study, with each subject serving as their own control. Subject-specific parameters could certainly change the results numerically but would not likely change the differences between conditions such that drastically different conclusions were reached. In addition, the probabilistic approach to fatigue modeling is in fact intended to account for a lack of subject-specific model parameters. By deriving the model's parameters from scattered experimental data, the probability produced for each subject is the estimated probability of failure, given uncertainty in the that specific subject's parameter values on joint structure, material properties, stress-life relationship, etc.

The overall conclusion suggested by these results is that contact mechanics and natural repair cannot explain why running does not appear to increase the risk of developing knee osteoarthritis, even with very generous assumptions on the capacity of these elements of cartilage to affect its long-term health: the natural repair capacity of cartilage has long been thought to be limited due to the typical isolation of cartilage from innervation and

vascularization (*Mankin, 1974*), and differences in cartilage modulus as a function of loading rates relevant to human locomotion are minimal (*Oloyede, Flachsmann & Broom, 1992*). The model could only produce the result seen in the epidemiology literature (no increase in knee osteoarthritis risk from running) when adaptation was included in the model. The present results therefore refute the "cumulative load" hypothesis on why running does not often cause osteoarthritis (*Miller et al., 2014*): even though the medial knee load accumulated per unit distance traveled was statistically equivalent in running vs. walking, replacing 3 km/day of walking with running increased the damage accumulated per unit distance and the probability of long-term structural failure (Fig. 7). Relatedly, the present results support the "cartilage conditioning" hypothesis on why running does not often cause osteoarthritis (*Seedhom, 2006*): running places high stresses on cartilage, and in a healthy state these stresses trigger an adaptation response that extends the fatigue-life of cartilage. The present results add the suggestion that bone adaptation indirectly extends cartilage fatigue life, independent of changes to the structure and function of cartilage itself: osteogenesis of new cortical bone in the spirit of Wolff's Law, modeled here as increases in the tibiofemoral radii of curvature, reduced cartilage damage and reduced the extent of adaptation needed by cartilage itself for long-term joint health.

A question motivated by this conclusion is whether the adaptations modeled here to cartilage elastic modulus, cartilage thickness, and tibiofemoral radii of curvature are plausible. When the parameters were adjusted in isolation, only the modulus had a value within the range of the referenced human cartilage data. Although most of these data were from elderly, deceased subjects who were not likely well-trained runners, the necessary adjustments to the other two parameters (+5.35 and +6.70 SD) are extreme enough to question their plausibility. However, when adjusting all three parameters at once, the failure probability for running could match that of walking with parameter values within the referenced data ranges for all three parameters. These between-subjects ranges do not necessarily indicate the plausible ranges of within-subject adaptations, but animal models provide evidence on the feasibility of the increases reported here, which ranged from 13–35% of the original values. After 15 weeks of running training in dogs, *Kiviranta et al. (1988)* reported 19–23% thicker cartilage (vs. controls), and *Jurvelin et al. (1986)* reported an average 10% greater elastic modulus and 28% greater femoral cartilage glycosaminoglycan content, which correlates with cartilage stiffness (*Samosky et al., 2005*). *Oettmeier et al. (1992)* reported an average 10% greater subchondral bone plate thickness (vs. controls) after 40 weeks of running training in dogs, and *Murray et al. (2001)* reported 57% greater thickness in a similar experiment on horses after 19 weeks of intense running training. Similar data on running training in humans are not available, and even these animal model studies are not repeated-measures designs (measurement of the outcome variables required euthanasia), but they suggest the adaptations modeled here are not implausible over years of consistent training. Adaptations of other elements of bone or cartilage that were not modeled here are also feasible, which would reduce the magnitude of necessary adaptations by these three parameters. Reduction or total removal of the repair capacity does not greatly influence these estimates, since repair had such a small effect on the failure probability when running (Fig. 7).

The modeled adaptation here to tibiofemoral radii of curvature may be particular difficult to justify. For example, the adaptation of +1.15 SD equates to an increase in subchondral tibia bone thickness of ∼40–60%, assuming a baseline thickness of ∼5 mm (*Oosthuyse et al., 2017*). Cross-sectionally, young runners had only about 18% thicker subchondral tibia bone vs. sedentary controls (*Oosthuyse et al., 2017*), but the animal studies cited above suggest longitudinal increases of ∼57% are plausible. It would be possible, for example, to increase the cartilage modulus by +1.50 SD, and require less increase in subchondral bone thickness to match the failure probability of walking. The relative adaptability of these parameters *in vivo* to running training are currently unclear.

Along with the animal training studies cited above, the plausibility of cartilage and bone adaptations can be roughly assessed in studies making cross-sectional comparisons between runners and non-runners. Male competitive masters runners (age 62 ± 12 years) had 6.5% greater tibia epiphyseal bone area compared to sedentary controls, but the same finding was not seen in female runners, and the epiphyseal bone mineral density did not differ between runners and controls (*Wilks et al., 2009*). Middle-aged marathon runners (age 53 ± 5 years) had 35% thicker tibiofemoral cartilage than sedentary controls, but the same finding was not seen in young (age 26 ± 5 years) marathon runners (*Mosher, Liu & Torok, 2010*). $T_1$-relaxation time in dGEMRIC, an index of glycosaminoglycan concentration which is thought to affect cartilage stiffness, was 15% greater in recreationally active individuals vs. sedentary controls, and 28% greater than controls in elite runners training an average of 90 km/week (*Tiderius et al., 2004*). These cross-sectional differences do not appear to generalize to all runners, cannot be confidently attributed to running-induced adaptation, and do not imply the magnitudes of adaptations modeled here are realistic. However, they suggest the notion of running inducing positive adaptations in bone and cartilage that benefit long-term cartilage health is worth further investigation.

## CONCLUSION

In conclusion, these model-based results suggest that (i) in the absence of repair and adaptation, running accumulates a great deal of damage per unit distance traveled, vs. walking the same distance, and (ii) sustaining a lifetime of mechanical loading from running without structural failure of the loaded cartilage is unlikely without an adaptation response induced by mechanosensitive cells to alter structure and function in a way that extends fatigue-life, i.e., cartilage conditioning.

### Funding

This work was supported by the DoD-VA Extremity Trauma & Amputation Center of Excellence (Public Law 110–417, National Defense Authorization Act 2009, Section 723), and the Center for Rehabilitation Sciences Research at the Uniformed Services University of Health Sciences (Principal Investigator: Paul F. Pasquina; DoD Defense Health Program

NF90UG). The funders had no role in study design, data collection and analysis, decision to publish, or preparation of the manuscript.

### Grant Disclosures
The following grant information was disclosed by the authors:

DoD-VA Extremity Trauma & Amputation Center of Excellence.

Center for Rehabilitation Sciences Research at the Uniformed Services University of Health Sciences.

DoD Defense Health Program: NF90UG.

### Competing Interests
The authors declare there are no competing interests.

### Author Contributions
- Ross H. Miller conceived and designed the experiments, performed the experiments, analyzed the data, prepared figures and/or tables, authored or reviewed drafts of the paper, created and performed computer simulations, and approved the final draft.
- Rebecca L. Krupenevich conceived and designed the experiments, performed the experiments, analyzed the data, authored or reviewed drafts of the paper, and approved the final draft.

### Human Ethics
The following information was supplied relating to ethical approvals (i.e., approving body and any reference numbers):

The University of Maryland Institutional Review Board granted ethical approval to carry out the study (IRBnet ID 671084).

### Data Availability
The gait data and Matlab codes for running the models reported in the article and reproducing the figures are available in Figshare: Miller, Ross; Krupenevich, Rebecca (2020): Data to accompany ''Medial knee cartilage is unlikely to withstand a lifetime of running without positive adaptation: a theoretical biomechanical model of failure phenomena''. figshare. Dataset. https://doi.org/10.6084/m9.figshare.12006312.v1.

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
