# Peer review of "Medial knee cartilage is unlikely to withstand a lifetime of running without positive adaptation: a theoretical biomechanical model of failure phenomena"

_PeerJ, doi:10.7717/peerj.9676_

## Round 0.1 · original submission · Major Revisions

· Academic Editor

Major Revisions

Your paper has received three enthusiastic reviews-- all see the value and all have constructive critiques that will improve it, but these are numerous, not merely stylistic, and will require re-review after substantial revisions. Well done on an ambitious and fascinating study- we look forward to the revised version.

Reviewer 1 ·

Basic reporting

no comment

Experimental design

no comment

Validity of the findings

no comment

Additional comments

The Authors present a study that aims to understand and predict cartilage failure as a function of time in a theoretical manner. They utilize state-of-the art modeling methods that combine motion (gait) analysis, musculoskeletal modeling of joint contact forces and discrete element modeling coupled with phenomenological modeling of material (here articular cartilage) failure. The reviewer commends authors on their approach which accounts for some limitations of previous approaches. However, the Reviewer does not fully agree with the conclusions that the authors present. Although, there are some issues that should be considered before this manuscript can be accepted for publication, the reviewer thinks that the results of this paper are very interesting.

Specific comments:

Introduction
Lines 64-67: Why authors want only to focus on modeling tissue fatigue/failure? This is not necessarily the best approach as this neglects the fact that cartilage has at least different constituents that can fail (collagen or proteoglycan matrix). This is actually very relevant aspect as it is not currently known which component damages first (and/or the initiation can be OA phenotype dependent). The reviewer suggests that authors should also account for and present briefly some of the recent modeling approaches that actually aim at simulating and predicting the progression of OA. In these approaches, the compositional properties of specific tissue constituents (collagen fibrils, proteoglycan matrix) are assumed to degenerate. Some good examples are e.g. Hosseini et al. 2014 (https://www.ncbi.nlm.nih.gov/pubmed/24185112), Mononen et al. 2018 (https://www.ncbi.nlm.nih.gov/pubmed/29150953) and Orozco et al. 2018 (https://www.ncbi.nlm.nih.gov/pubmed/30348953). It would also be beneficial to discuss the differences bwtween the approaches presented by the authors and these papers in the discussion section.

Methods:

This is a suggestion but the reviewer would prefer a general overview/methodological figure which illustrates the whole workflow and the geometries used. The reviewer finds it hard to think how the geometry of cartilage and meniscus looks in the model when seeing the figure 1.

Lines 130-136: Authors use non-linear elastic constitutive relationship (which is apparently a function of the logarithmic (or true) strain), however it seems that this relationship does not account for any rate-dependency (i.e. poro or viscoelastic behavior) so this reviewer thinks that saying this stress-strain relationship accounts for the rapid deformations is incorrect (although it is better suitable for large deformations as the authors mention).

What is composite modulus in your model? I think it is better to say dynamic/instantaneous/equilibrium modulus of cartilage/meniscus as this is what you are modeling and it avoids confusion.

Also, it is not clear why you have two indexes (i and j) for the number of elements. Further, I guess these indexes actually refer to the stress and strain matrix indexes, and this confuses the reviewer. I suggest revising the equations for clarity.

Lines 158-168: The authors fit Weibull cumulative probability function on data presented in Reimenschneider et al. 2019. As this fit is fundamental part of your paper, it would be good to mention R2-value and plot the resulting fit (with original data). I tried to plot the e.q. 4 and the result seemed odd, so this would be good to verification (it is also possible that the reviewer made a mistake). How unique was your fit (i.e. by varying initial values did you receive a similar fit and parameters) and what approach was used for fitting?

Also it is not clear for the reviewer if is there a "feedback" i.e. is modulus change if damage is occurring? If this is not present, it should be mentioned as a limitation as it would present a different course in the progression (most likely faster "complete" failure). And how was the adaptation actually incorporated, immediately in the initial material properties of the model or during the analysis?

Methods (Implementation and statistics): It is not clear for the reviewer how the motion was simulated as a function of time. Did you simulate one full gait cycle (or just stance phase) several times?

Discussion:
Lines 311-313: This sentence is difficult to grasp, could this be revised?

Lines 335-336: I guess it would be also good to mention that one (quite big) limitation is that authors neglect poro/viscoelastic properties of cartilage and meniscus that contribute to the rate dependency of the deformation.

Lines 352-363: The statements that the authors present is true only for the modeling approach that authors are using, and this should be clearly said.

Authors also mention that "running places high stresses on cartilage, and in a
358 healthy state these stresses trigger an adaptation response that extends the fatigue-life of cartilage". Although this might be correct, I am quite certain that this is not necessarily so simple as the fluid pressurization is neglected in the model. What were the estimated strains in cartilage (on average) during walking vs. running in these subjects, same or different? If fluid pressurization would be included I would assume that with otherwise similar material properties (lets assume, for simplicity, isotropic linear elastic) during running loading rate is faster and, thus, fluid pressurizes more (and fluid pressure inside tissue is greater) compared to walking, which could be harmful for cells (or even cause apoptosis) in the tissue or damage collagen network.

Further, I find it very challenging to accept that tibiofemoral radii would increase/decrease so substantially as the authors present (was this a change that was observed in running vs. walking people). Further, subchondral bone can remodel and adapt but I would assume that the bone mineral density (and thus modulus) is changing based on the over-/under loading of the cartilage and subchondral bone.

·

Basic reporting

The writing was clear and well structured. The figures were clear and easy to interpret.

The literature on the short-term training effects of running on cartilage was well summarized. However, no literature was mentioned that cross-sectionally compares runners to non-runners in terms of anatomy or bone densitiy which may give some understanding to the long-term adaptations that are relevant to OA.

Experimental design

Generally the study was well designed with clear hypotheses and subsequent evaluation.

Validity of the findings

The conclusions are adequately supported by the computational experiments performed in the study. The study obviously lacks experimental validation aside from a minimal comparison to epidemiological rates of OA, however this limitation is clearly stated in the discussion.

Additional comments

G1) The first hypothesis is based on contact stress (no difference between peak stresses in walking and running), whereas the second hypothesis is based on contact strain (no difference between damage in walking and running, where damage is a function of strain, Eq 4 & 5). The adaptations used to test the third hypothesis are evaluated based on damage (strain), but some would have opposite effects for stress (increasing elastic modulus decreases strain, but increases stress for a given input contact force). A better discussion on whether cartilage stress or strain is the important metric to OA development, and more explicit description that the damage model is based on strain could help clarify the reasoning behind your hypotheses.

G2) When comparing cartilage loading mechanics between walking and running, I would expect the viscoelastic properties of the tissue to be important due to the impact loading during running. The complex fluid-solid interactions that occur within the extracellular matrix ultimately determine the chondrocytes mechanical environment and biologic response [1]. Some justification should be provided that the joint contact model is of high enough fidelity to address the hypotheses being tested.
[1] Halloran, J. P., Sibole, S., Van Donkelaar, C. C., Van Turnhout, M. C., Oomens, C. W. J., Weiss, J. A., ... & Erdemir, A. (2012). Multiscale mechanics of articular cartilage: potentials and challenges of coupling musculoskeletal, joint, and microscale computational models. Annals of biomedical engineering, 40(11), 2456-2474.

G3) It appears that the geometry of the contact model does not scale with subject size, and that the absolute medial contact forces are applied to the joint contact model. It is generally assumed that joint contact forces should be compared across subjects by normalizing to body weight. In your method, if runner A weighs twice runner B, but they run with a similar peak knee contact force of 4x BW, you would apply double the force to the same geometry in the contact model, and thus expect double the stresses.

·

Basic reporting

The manuscript reads very well and structured very well. It meets PeerJ and any scientific journal standards.

Experimental design

1) Page 7 Lines 92-93: Maybe a little more information on the capture marker set, kinematics and inverse dynamics modelling is desirable.

2) Methods: A data and modelling work flow chart might also be good.

3) Page 7 Lines 113-116: Please provide a plot of the model estimated medial-lateral force with those from instrumented implants: I suggest doing this on your current figure 2. Please let the reader make a decision about the performance of the NMSK knee joint model.

4) Page 8 Section 2.5.1: It would be nice to see how well the damage model fits Riemenschneider et al. (2019) data on a figure. Maybe this could be done in supplementary files. This would also give the reader a feel for the damage model being used.

5) Section Damage and Repair models:

a. Rate of loading will affect the material properties since cartilage is poroviscoelastic structure. Running has a higher loading rate than walking, resulting in stiffer cartilage response, i.e. changing in the dynamic cartilage elastic modulus (see for example, Nia et al. J Biomech 48, 2015, 162–165). How will this affect Damage and Repair as modelled?

b. It would be good to see some figures showing how the Damage and Repair performs in isolation of the walking or running loading. Are there any experimental data that can be used for benchmarking?

6) Page 10 Lines 224-225: Given that poroviscoelastic effects on cartilage dynamic elastic modulus and the effects on loss and increase in GAG’s, with the associated effects on cartilage elastic modulus, are not cartilage strains more important than stress and hence the third hypothesis is possibly different?

Validity of the findings

1) Page 11, line 258-259: I suggest change “Model validity was…” to “The validity of various sub-model components was…” Also see comments 4 and 5 in in Experimental design section: showing the performance of Damage and Repair models with literature would be beneficial.

2) Page 11, Lines 268-269: How do you running loads compare to those reported in literature.

3) Page 11, Lines 257-273: Peak knee loads from yours and other models may occur at different timings and knee flex-extension angles. Knee angles is important since this means a different region of the cartilage is loaded.

4) Page 11, lines 274-275: Th first sentence of this paragraph is a Figure caption. I suggest rewrite to “The time to estimated medial tibiofemoral cartilage failure depended on how much walking and running was undertaken and the inclusion of repair (Fig. 4).” Or something similar.

5) Page 11, lines 274-289: Rate of loading will affect the material properties since cartilage is poroviscoelastic structure. Running has a higher loading rate than walking, resulting in stiffer cartilage response. How will this affect the results given the relative mix of walking and running is examined, i.e. Fig. 4.?

6) Page 12, Discussion, First Hypothesis: Focus on stress rather than strain – see comments 5 and 6 in Experimental design section.

7) Page 12, Line 313: I think this should be Fig 5. not Fig 4.

8) Page 13, Lines 364-387: The fact of dynamic elastic cartilage modulus, and its possible differences between walking and running could also account for reduced risk of cartilage damage in running. This fact may not require a repair model to be included – please comment.

9) Fatigue model of damage: There are other proposed models of cartilage damage. For example, cartilage has hydrodynamic lubrication from fluid exudation which lowers friction and cartilage-cartilage contact preventing shearing off of cartilage surfaces.

Additional comments

Generally, this manuscript is where analyses should be taken to understand the development of knee OA, and it will add to the literature. I really enjoyed reading it. However, I encourage the authors to address the above comments.

---

## Round 0.2 · accepted · Accept

· Academic Editor

Accept

One of the two reviewers was available and they are very pleased with revisions so I am happy to provisionally accept this-- congratulations!

Reviewer 1 ·

Basic reporting

-

Experimental design

-

Validity of the findings

-

Additional comments

The authors have done excellent job in their response and revision. I have no more comments. I am pleased to see this paper to be published in PeerJ.